# The Relationship between Plasma Taurine Levels and Diabetic Complications in Patients with Type 2 Diabetes Mellitus

**DOI:** 10.3390/biom9030096

**Published:** 2019-03-11

**Authors:** Duygu Sak, Fusun Erdenen, Cuneyt Müderrisoglu, Esma Altunoglu, Volkan Sozer, Hulya Gungel, Pınar Akca Guler, Tuncer Sak, Hafize Uzun

**Affiliations:** 1Department of Internal Medicine, Istanbul Training and Research Hospital, Fatih, Istanbul, 34098, Turkey; duygu_duygucuk@hotmail.com (D.S.), fusunozerdenen@hotmail.com (F.E.), cuneytmuderrisoglu@gmail.com (C.M.), esmaaltunoglu@yahoo.com (E.A.), dr.tuncersak@gmail.com (T.S.); 2Department of Biochemistry, Yildiz Technical University, Esenler, Istanbul, 34220, Turkey; volkansozer2000@yahoo.com; 3Department of Ophthalmology, Istanbul Training and Research Hospital, Fatih, Istanbul, 34098, Turkey; hulyagungel@msn.com (H.G.), pnr_akca@hotmail.com (P.A.G.); 4Department of Biochemistry, Cerrahpasa Faculty of Medicine, Istanbul University-Cerrahpasa, Fatih, Istanbul, 34098, Turkey

**Keywords:** diabetes, taurine, diabetic retinopathy, diabetic neuropathy, microalbuminuria

## Abstract

*Background:* Taurine has an active role in providing glucose homeostasis and diabetes causes a decline in taurine levels. This paper investigates the relationship between taurine and diabetic complications, patients’ demographic features, and biochemical parameters. *Methods:* Fifty-nine patients with type 2 diabetes mellitus (T2DM), and 28 healthy control subjects between the ages of 32 and 82 were included in the study. The mean age of subjects was 55.6 ± 10.3 and mean diabetes duration was 10.2 ± 6.0 years. The most commonly accompanying comorbidity was hypertension (HT) (64.5%, *n* = 38), and the most frequent diabetic complication was neuropathy (50.8%, *n* = 30). Plasma taurine concentrations were measured by an enzyme-linked immunoassay (ELISA) kit. *Results:* Plasma taurine concentrations were significantly lower in diabetic patients (0.6 ± 0.1 mmol/L) than controls (0.8 ± 0.2 mmol/L) and in hypertensive (0. 6 ± 0.1 mmol/L) patients (*p* = 0.000, *p* = 0.027 respectively). *Conclusion:* Plasma taurine levels were decreased in patients with T2DM and this was not related to FBG, HbA1c, and microalbuminuria. With regard to complications, we only found a correlation with neuropathy. We suggest that taurine levels may be more important in the development of diabetes; however, it may also have importance for the progression of the disease and the subsequent complications. We further assert that taurine measurement at different times may highlight whether there is a causal relationship in the development of complications.

## 1. Introduction

According to the World Health Organization (WHO) reports, there are 415 million patients with diabetes mellitus (DM) and 318 million with impaired glucose tolerance (IGT) [1,2]. Based on the TURDEP-II study, type 2 DM incidence reaches up to 13% in Turkey [3]. DM and its complications worsen patients’ quality of life and are a huge burden to economies. The two major risk factors for developing complications are diabetes duration and chronic hyperglycemia. Other factors are bad glycemic control, puberty, pregnancy, hypertension (HT), impaired lipid profile, anemia, the interaction within some mediators like growth factors, nitric oxide, angiotensine-2, endotheline and changes in the microcirculation, increased polyol pathway activity, glycation end products, and oxidative stress [4].

Taurine was first isolated from bile by Friedrich Tiedemann and Leopold Gmelin in the early 1800s [5]. Taurine (2-aminoethanosulphonic acid) is an essential amino acid which is found much in mammalian tissues, mother’s milk, and sea products. This methionine and cysteine-derivated amino acid contains sulphonate but not the carboxyl group and is primarily synthesized by oxidation and decarboxylation of cysteine in the liver [6,7]. Although its biosynthetic capacity is low in the human body, oral intake is possible. Up to 70 g taurine may be found in a 70 kg person’s body. It is found in high concentrations in secretory tissues, platelets, and in tissues with high electrical activity like the brain, heart, and retina [8]. Its physiologic roles can be line up as follows: It serves as a substrate for conjugation reactions of bile acids in the liver, has a role in stabilizations of membranes, calcium homeostasis, osmoregulation, cytoprotection, in development of brain and retina; it controls neurotransmitter secretions, regulates secretions of proinflammatory cytokines and lowers blood pressure and it has antioxidant properties [5,9]. Taurine also plays an effective role in glucose homeostasis, but the specific molecular mechanisms of this role are unknown. Taurine exerts effects in glucose homeostasis through two known mechanisms: i) By its effects upon β-cell insulin secretion. ii) By interfering with the insulin signaling pathway and post-receptor events. [7,10,11]. Its deficiency can cause metabolic impairment and is related to the development of nearly all diabetic complications [5,6]. Animal and human models show that taurine supplementation decreased the incidence of some disorders like DM, HT, hyperlipidemia, and obesity [9,12].

Taurine deficiency is linked to diabetic complications. To the best of the authors’ knowledge, this is the first study evaluating the relationship between plasma taurine levels with diabetic retinopathy (DR), neuropathy (DN), and nephropathy. We investigated the relationship between plasma taurine levels and T2DM, and its complications and patient demographic and biochemical parameters.

## 2. Materials and Methods

### 2.1. Subjects

Fifty-nine patients with type 2 diabetes and 28 healthy control subjects between the ages of 32 and 82 were included in the study. Subjects who were admitted to the Istanbul Training and Research Hospital were included. All participants were informed about the study and they signed an informed consent form. The protocol was approved by the local Ethics Committee of the Istanbul Education and Research Hospital (no:793, date:11.03.2016) and was conducted in accordance with the Declaration of Helsinki.

Pregnant women, patients with acute infection, thyroid disorders or acute vascular accident, or a history of malignancies were excluded. Patients with T2DM were followed up with at the Outpatient Clinic of Diabetes and were diagnosed according to ADA criteria. Hypertension (HT) was accepted as having systolic blood pressure (SBP) > 130 mm Hg and/or diastolic blood pressure (DBP) > 85 mmHg or antihypertensive drug usage [13]. Height, weight, and waist circumference (WC) were measured by the same person; body mass index (BMI) was calculated. DN was diagnosed according to a questionnaire and physical examination to determine thermal and mechanical sensitivities. We could perform electromyography (EMG) in some patients. DR diagnosis was made by ophthalmologic examination and angiographic findings. Spot urine microalbumin level was accepted as a marker of nephropathy. We also measured serum creatinine levels and glomerular filtration rate (GFR) of the subjects. Patient history, electrocardiography (ECG), and further investigations according to the patient’s needs were performed for coronary heart disease (CHD). Lastly diabetic foot (DF) was diagnosed by history and physical examination.

### 2.2. Blood Sample Processing

All subjects had at least 12 hours of fasting before blood sampling for biochemical analysis. Blood samples were processed in a centrifuge at 3000 rpm. Plasma was collected and stored at −80 °C until analysis. Fasting blood glucose (FBG), creatinine, HbA1c, insulin, C-peptide, lipid profile, and urinary microalbumin concentrations were measured spectrophotometrically using the Abbot Aeroset 2.0 (Abbot Diagnostic, Illinois, USA). HbAIc (%) was measured by the Premier Hb9210 (Trinity Biotech, Wicklow, Ireland) which uses the glycation specific binding of boronated affinity to detect all the glycated Hb species present. Estimated glomerular filtration rate (eGFR) was performed according to the Chronic Kidney Disease Epidemiology Collaboration (CKD-EPI) equation.

### 2.3. Measurement of Plasma Taurine Levels

Plasma taurine levels were measured by a commercially available competitive enzyme-linked immunoassay (ELISA) kit (Mybiosource, Catalog Number: MBS756338, Inc. San Diego, CA, USA). The coefficients of intra and inter-assay variation were 4.7% (*n* = 15) and 5.9% (*n* = 15), respectively.

### 2.4. Statistical Analysis

Mean value, standard deviation, median, minimum, maximum, and frequency were used to describe the data statistically. Statistical analyses were performed using the SPSS 22.0 Programme (NCSS, LLC 329 North 1000 East Kaysville, Utah, USA). The distribution of variables was investigated using the Kolmogorov–Smirnov test. For quantitative analysis ANOVA, Kruskal–Wallis, Mann–Whitney U tests were used. A chi-square test for the comparison of qualitative variables was used when possible; when this test was not suitable we used Fischer test. For numerical data, the Spearman Correlation test was used to see the relationship of variables. One-way variance analysis for comparison of groups and Tukey test for subgroup comparisons were also used. Values of *p* < 0.05 were considered to be statistically significant with a 95% confidence interval.

## 3. Results

Demographic, clinical, and laboratory features of control and diabetic groups are shown in Table 1. A total of 87 subjects, with a mean of 55.6 ± 10.3 years, were enrolled in the study, 47 female and 40 male. The mean diabetes duration of the population was 10.2 ± 6.0 years. The most common comorbidity was HT (64.5%, *n* = 38) and the most common diabetic complication was neuropathy (50.8%, *n* = 30). Plasma taurine concentrations were significantly lower in diabetic patients (0.6 ± 0.1 mmol/L) than controls (0.8 ± 0.2 mmol/L) and in hypertensive (0.6 ± 0.1 mmol/L) patients (*p* = 0.000, *p* = 0.027 respectively). Diabetes duration was 10.8, 11.2, 11.4, and 11.6 years for complications (retinopathy, neuropathy, nephropathy, and CHD respectively). There was no difference between plasma taurine levels of female (0.70 ± 0.23) and male (0.68 ± 0.20) subjects (*p* = 0.95). Demographic, clinical and laboratory characteristics of the control and diabetic groups are shown in Table 1.

Diabetic patients were older and more hypertensive than controls. As expected FBG, HbA1c, C-peptide levels were different between control and diabetic subjects; whereas HDL, LDL, triglyceride, urea, uric acid, and fibrinogen levels were similar. Microalbuminuria was significantly higher and GFR and Taurine levels were significantly lower in DM group.

There was no correlation between taurine concentrations and age, BMI, WC, HC, SBP, DBP, diabetes duration in the diabetic group. There was not a correlation between taurine levels with FBG, HbA1c, microalbuminuria, insulin, C-peptide, creatinine and GFR values in the diabetic group.

We accepted microalbuminuria as a marker of diabetic nephropathy because 24 of 59 diabetic patients had micro and macroalbuminuria, whereas few subjects had high creatinine levels. There were 24 patients with GFR values under 90 mL/min, whereas 9 patients had creatinine levels higher than 1 mg/dL. When we accepted GFR as an indicator of diabetic nephropathy, we could not find a correlation with GFR values and taurine levels.

With regard to the association of taurine levels with comorbid HT and complications in the diabetic group, results are shown in Figure 1. The only correlation was found between serum taurine level and the DN (*p* = 0.036). With respect to comorbid hypertension and taurine association, in the nonhypertensive diabetics, hypertensive diabetics and nondiabetic hypertensive groups, taurine levels were 0.56 mmol/L, 0.62 mmol/L and 0.63 mmol/L, respectively.

## 4. Discussion

In our study, the taurine concentration of the control group was significantly higher than that of the DM group. Taurine concentrations were significantly lower in patients with DN without showing an association with other diabetic complications.

Merheb et al. [14] showed that plasma taurine concentrations were significantly lower than the control group, similar to our study, and they also stated that taurine supplementation can prevent the development of DM and its complications. Another study by Copeland et al. [15] showed that placental taurine levels were decreased in pregnant rats which were made DM via streptozotocin injection. Taurine is suggested to be necessary for beta cell function [16]. In our study, plasma taurine levels were higher in the control group than in the DM group in accordance with the literature. Some studies, conducted on obese or diabetic rats, showed that taurine supplementation ameliorated glucose tolerance and insulin sensitivity along with decreasing insulin resistance, hyperinsulinemia, and hyperglycemia [9,11,17,18]. Vettorazzi et al. [19] revealed that taurine increased glucose-dependent insulin release. In the light of these findings, it is expected that plasma taurine levels and hyperglycemic state are inversely correlated to each other. In our study, we found that the group which had high taurine levels had low FBG.

Taurine is known to play an effective role in glucose homeostasis, and its deficiency can cause metabolic impairment and the development of diabetic complications [5,6]. The diabetic patients were receiving treatment so their plasma glucose values might not reflect the real association. Some of our diabetic subjects were receiving only oral antidiabetic drugs, some only insulin, and the others both treatments. To avoid small groups, we did not evaluate the relation of taurine levels with treatment type. Future studies which evaluate subjects at the time of diagnosis and at different times will provide more valuable and significant results about the associations discussed.

Taurine insufficiency may be a risk factor for diabetic nephropathy [20]. When given in proper doses, taurine supplementation has a beneficial effect on DM and diabetic complications as a coadjuvant therapy [5,6,12]. Preventive effects might be due to anti-hyperglycemic, antioxidant, antifibrotic, and anti-inflammatory properties [6,9,21]. A study by Trachtman et al. [22] showed that taurine supplementation had decreased albuminuria by about 50%. This effect was thought to be related to the decreased lipid peroxidation and accumulated glycosylation end products in renal parenchyma [22,23]. Taurine may increase the effect of metformin and using them together could ameliorate the loss of renal function, renal oxidative stress, and impaired lipid and glucose metabolism [24]. Koh et al. [25] showed that the diabetic rats which were treated with taurine had decreased microalbuminuria and glomerular mass and increased pore density and glomerular basement membrane thickness. In our study, when micro- or macroalbuminuria was taken as an indicator of diabetic nephropathy, patients with nephropathy had lower levels of taurin without a statistically significant relationship with plasma taurine levels. We suppose that this may be due to the renal protective effect as previously mentioned. Diabetic patients were on antidiabetic treatment and patients were taking angiotensin-converting enzyme inhibitors (ACEI) or angiotensin receptor blockers (ARB). This treatment may prevent excessive albuminuria.

There are high concentrations of taurine in neural and nonneural tissues of the eye. Retinal taurine plays an important role in the development of photoreceptors and stress-related neural damage [26]. Streptozotocin-induced diabetic rats had decreased taurine concentrations in retinal pigment epithelium and photoreceptors. Taurine has a protective effect against hypoxia-related retinal ganglion damage; it also prevents diabetes-related apoptosis of retinal cells [27,28,29,30]. We did not find a correlation between plasma taurine and retinopathy in the diabetic group in contrast with literature.

It was determined that taurine deficiency was related to increased neural damage and instability [31,32]. Furthermore, it was concluded that taurine deficiency aggravated neural damage, slowed the speed of neural transmission. Conversely, taurine supplementation decreased oxidative stress and accelerated the speed of neural transmission. Taurine supplementation also ameliorated endoneurial blood circulation [33]. In our study, we observed that plasma taurine levels were significantly lower in neuropathic patients than both DM patients without DN and the control group. This inverse correlation was thought to be due to the positive effect of taurine in the regulation of diabetes and the restoration of the central and peripheral nervous system. We also suggest that taurine supplementation may be beneficial in protecting against and the treatment of neuropathy. Epidemiological studies showed that diabetic peripheral neuropathy ranges from 5% to 100%, the variation may be due to the differences in the study populations and the diagnostic methods. Evidence of neuropathy was found to be 7.5% at the time of diabetes diagnosis [34]. We evaluated our DR patients thoroughly by fluorescein angiography. We think that this is the reason why DR is the top complication, considering diabetes duration in our patients. EMG and coronary angiographies were not performed on all subjects, and, therefore, some DN and CHD patients may have been missed.

Several epidemiological and experimental studies showed that dietary taurine lowers the blood pressure [35,36,37]. Taurine shows its antihypertensive effect by suppressing the renin-angiotensin-aldosterone (RAS) system and stimulating the renal kinine kallikrein system. Taurine also decreases the cardiac hypertrophic effect of angiotensin. It inhibits sympathetic activity and dilates vessels which are involved in hypotensive effects [35]. There was an inverse correlation between plasma taurine concentrations and SBP in all subjects in our research. Furthermore, patients with a history of HT had significantly lower plasma taurine than patients who did not have HT. With regard to comorbid HT in diabetics, there was no difference in plasma taurine levels in hypertensive and non-hypertensive subjects.

In a study on middle-aged otherwise healthy females, taurine supplementation had protective effects against cardiovascular disease (CVD). Researchers suggested this effect might be via decreased homocysteine levels, which is an independent risk factor for CVD [6,35,38]. Taurine may also have a positive effect on cardiac contractile functions and structural changes [39]. Urinary taurine excretion was also inversely correlated with cardiac mortality [16]. In our study, we did not find any correlation between plasma taurine and CHD. However, CHD was diagnosed according to history and ECG findings, and, thus, some cases might be omitted.

There are many limitations to our study. First we had a limited number of patients; diabetic and nondiabetic group were not matched in terms of age and gender; the patients were already on treatment; factors that may alter plasma taurine concentrations like obesity, hyperlipidemia, or nutritional status, etc. were not considered; measurement of taurine levels was made only once; gold standard angiographic evaluation was not used to establish CHD. For ethical reasons, we could not perform a washout period for patients. We did not include newly diagnosed diabetic patients either and this may affect our results due to the medications which the diabetic subjects used.

## 5. Conclusion

In this study, we found an inverse correlation between taurine and the parameters used to diagnose and follow up T2DM like FBG, HbA1c, and albuminuria. With regard to complications, we only found a correlation with neuropathy. In our opinion, this study, despite failing to reveal a significant association between plasma taurine levels and diabetic complications, emphasizes the importance of taurine as a predictor of diabetic complications and even suggests an opportunity for its use in co-adjuvant therapy. We suggest that taurine levels may be more important in the development of diabetes but become less important after the progression of diabetes and its complications. In light of these findings, we think taurine as a marker can be used to predict the progression of diabetes in obese subjects with risk. In addition, it is possible to use taurine for treatment in the future.

## Figures and Tables

**Figure 1 biomolecules-09-00096-f001:**
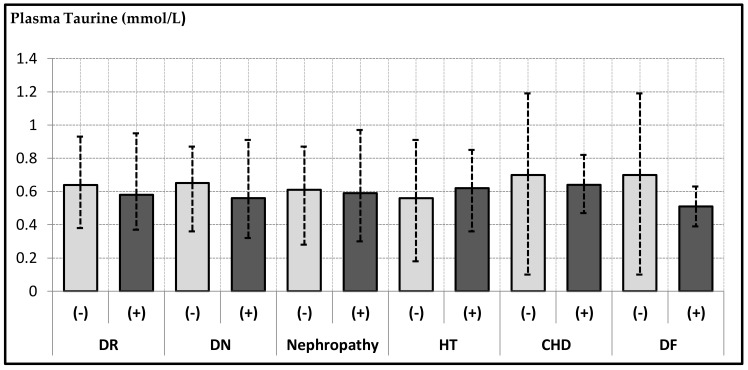
Relationship of plasma taurine levels with diabetic complications in diabetic subjects. DR: diabetic retinopathy; DN: diabetic neuropathy; HT: hypertension; CHD: coronary heart disease; DF: diabetic foot.

**Table 1 biomolecules-09-00096-t001:** Demographic, clinical and laboratory features of control and diabetic groups.

Diabetic Group				Control Group			
Gender							
Male	28			12			
Female	31			16			
	**Min**	**Max**	**Mean ± SD**	**Min**	**Max**	**Mean ± SD**	***p***
Age (years)	37	82	58.6 ± 9.2	32	77	48.3 ± 8.9	**0.001**
Height (cm)	133	185	161.8 ± 9.9	138	180	164 ± 10.6	0.346
Weight (kg)	52	115	79.3 ± 13.8	60	117	81.5 ± 15.2	0.508
BMI (kg/m^2^)	22.2	44.9	30.4 ± 5.6	22.1	41	29.8 ± 4.7	0.596
WC (cm)	70	130	96.1 ± 12	70	119	92.8 ± 13.1	0.245
HC (cm)	80	125	101.7 ± 9.8	90	130	102.1 ± 10.8	0.853
SBP (mm × Hg)	100	180	127 ± 15.3	100	160	125 ± 10	**0.001**
DBP (mm × Hg)	60	100	75.8 ± 8.1	60	90	74.6 ± 7.1	**0.007**
FBG (mmol/L)	4.38	24.64	10.16 ± 4.68	4.16	6.44	5.20 ± 0.63	**0.001**
HbA1c (%)	4.6	14.6	8.4 ± 2.1	5	6.5	5.7 ± 0.3	**0.001**
Microalbuminuria (mg/dL)	0.6	2367	209 ± 444.1	2	147	14.3 ± 28.4	**0.023**
GFR (mL/min)	8.56	118.89	86.05 ± 25.71	90.99	120.14	104.69 ± 8.05	**0.001**
Taurine (mmol/L)	0.2	0.9	0.60 ± 0.15	0.5	1.2	0.87 ± 0.22	**0.001**

BMI: body mass index; WC: waist circumference; HC: hip circumference; SBP: systolic blood pressure; DBP: diastolic blood pressure. FBG: fasting blood glucose GFR: glomerular filtration rate. Bold numbers are statistically significant.

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
