# Peer review of "The Relationship between Plasma Taurine Levels and Diabetic Complications in Patients with Type 2 Diabetes Mellitus"

_biomolecules, 2019, doi:10.3390/biom9030096_

Reviewer 1 Report

The Relationship Between Plasma Taurine Levels and Diabetic Complications in Patients with Type 2 Diabetes Mellitus.

I suggest the authors present the data more clearly with the aid of box plots and/histograms that can be more clearly

visualized and understood. The tables are not very clear and does not show the distribution of the data in the population considered.

The text has numerous spell and grammatical errors which makes it difficult to read through manuscript.

Results:

Table1: Data presented separately for diabetic and control subjects instead of pooling them would be helpful to understand the

association between different variables and markers better.

Table 2: if n= number of subjects, the total number of diabetic subjects used in the study (methods Ln 84) were 59. However, in Table 2,

number of subjects without DR considered are 60. Likewise, number of CHD, CVA diabetic subjects considered are 78 and 87 respectively.

Are these values correct?

Table 3: Do WC, HC, SBP and DBP qualify as demographic parameters? Have the authors compared taurine levels to gender?

Table 4: Please check the P values.

Table 5: Please provide the N numbers for each group. What was the duration of each complication in these subjects. How many of these

subjects also have other co-morbidities? For instance, did the authors analyze the correlation of plasma taurine levels in subjects that

have diabetes and SBP. As the authors have noted that patients being on medications as one of the  limitation of this  study, did the authors

find any correlation of plasma taurine levels with the type of medicine the patients were taking.

Table 6: Please provide the N number for each group. The authors don’t see a correlation of   levels with diabetic complications within

diabetic group except for neuropathy. What was the criteria for establishing neuropathy. Please provide more details.  What are the

other parameters that the authors consider for evaluating nephropathy besides albuminuria. Did the authors analyze the results with

creatinine levels. Was similar analysis done in control group alone.

Methods: Did the authors use established methods to measure biomarkers and clinical tests. What was the criteria for establishing Neuropathy. Please provide more details.

Introduction:

Line: Taurine also plays an effective role in glucose homeostasis; Please describe or briefly mention how.

Abstract: The abstract in the results section does not have details about blood glucose/ HbA1C / microalbuminuria/Nephropathy measures, However, they have included them in the results.

Author Response

Dear Editor,

First, we would like thank the reviewers for the helpful comments, which led us to conduct appropriate experiments.

The manuscript has subsequently been rewritten to address these concerns and comments of the reviewers.

We are grateful for your understanding and cooperation in this matter.

English

We have had our revised manuscript edited and proofread by a professional English-speaking editor (MDPI English editing) to

improve the readability and correct grammatical errors. We look forward to your reply. We believe that the language is now suitable for review.

Comments and Suggestions for Author:

The Relationship Between Plasma Taurine Levels and Diabetic Complications in Patients with Type 2 Diabetes Mellitus.

I suggest the authors present the data more clearly with the aid of box plots and/histograms that can be more clearly visualized and understood. The tables are not very clear and does not show the distribution of the data in the population considered. The text has numerous spell and grammatical errors which makes it difficult to read throughmanuscript.

Tables were corrected.

We have had our revised manuscript edited and proofread by a professional English-speaking editor (MDPI English editing) to

improve the readability and correct grammatical errors.

Results:

Table1: Data presented separately for diabetic and control subjects instead of pooling them would be helpful to understand the

association between different variables and markers better.

We changed Table 1 and separated the control and diabetic groups.

Table 2: if n= number of subjects, the total number of diabetic subjects used in the study (methods Ln 84) were 59. However, in Table 2, number of

subjects without DR considered are 60. Likewise, number of CHD, CVA diabetic subjects considered is 78 and 87 respectively. Are these values correct?

Table 2 was removed and the data was added to Table 4.  Numbers were controlled. 

Table 3: Do WC, HC, SBP and DBP qualify as demographic parameters? Have the authors compared taurine levels to gender?

We changed the heading of the Table 3 and also compared taurine levels to gender and wrote it in the result. Table 3 was changed to Table 2.

Table 4: Please check the P values.

We checked the P values.

Table 5: Please provide the N numbers for each group. What was the duration of each complication in these subjects. How many of these

subjects also have other co-morbidities? For instance, did the authors analyze the correlation of plasma taurine levels in subjects that have diabetes and SBP.

As the authors have noted that patients being on medications as one of the  limitation of this  study, did the authors find any correlation of plasma taurine levels

with the type of medicine the patients were taking.

We provided the N numbers for each group. The most common comorbidity was hypertension as mentioned. There were some patients

with gastrointestinal diseases such as peptic ulcus or irritable colon syndrome, hypothyroidism and anemia in diabetic group. We did not

document them. As seen in the result section patients with some disorders were excluded. We also presented taurin levels of diabetic patients

with hypertension and without HT. We did not make subgroups according to the drugs patients used. With regard to antidiabetic treatment

some were using OAD, some were on insulin treatment and the others were using both. The groups would be small, so we did not evaluate

the association of drugs with taurine level.

Table 6: Please provide the N number for each group. The authors don’t see a correlation of   levels with diabetic complications within diabetic group

except for neuropathy. What was the criteria for establishing neuropathy. Please provide more details.  What are the other parameters that the authors consider

for evaluating nephropathy besides albuminuria. Did the authors analyze the results with creatinine levels? Was similar analysis done in control group alone.

We provided the N numbers. Our criteria for establishing neuropathy was patient history. Most of the patients also had EMG evaluation.

As we mentioned in the method section we think that history and examination may be enough for diagnosis. But we also mentioned

that we might have missed some DN patients.  We accepted microalbuminuria as marker of diabetic nephropathy. We did not evaluate

the association of taurine level with serum creatinin and glomerular filtration rate. With regard to creatinin revel no relation was found

with its level and taurine in the whole subjects.  We wrote that laboratory results were similar between

Methods: Did the authors use established methods to measure biomarkers and clinical tests. What were the criteria for establishing Neuropathy? Please provide more details.

We used established biomarkers and clinical tests as decribided in the method section. Introduction:

Line: Taurine also plays an effective role in glucose homeostasis; please describe or briefly mention how.

We wrote known mechanisms about role of taurine in glucose homeostasis

Abstract: The abstract in the results section does not have details about blood glucose/ HbA1C / microalbuminuria/Nephropathy measures, However, they have included them in the results. .

We wrote details about blood glucose, HbA1c, microalbuminuria in the abstract.

Reviewer 2 Report

In their manuscript, the authors describe results of plasma taurine measurement in a group of patients with T2DM and in control

non-diabetic subjects. They also evaluate relationship between plasma taurine and diabetic complications.

The manuscript has several flaws:

1.       The description of subjects involved in the study is insufficient. Although the limitations are mentioned in the discussion,

subjects’ description must be improved. Characteristics of the subjects are shown in Table 1, however the authors pooled both groups and

showed the description for all patients. So the reader does not know which parameters differ between the groups. Furthermore, they

show minimum, maximum, median, mean and standard deviation which is redundant. I suggest using median and interquartile range as

not all parameters have normal distribution. Table 1 is poorly arranged, parameters should be aligned to the left and units should be

standardized (glucose is expressed in mg/dl and taurine in mmol/l). In the abstract, the authors show median but together with SD

which is incorrect. The reference to the Table 1 belongs in “2.1 Subjects” not in the results section. Patients should be better characterised

with respect to diabetic complications (e.g. CKD stage, GFR values, DR stage etc.).

2.       The authors mention lower plasma taurine in diabetic patients however this information (and respective P value) is missing in the results.

3.       Comparison of plasma taurine levels between patients with and without diabetes performed in all study subjects (including healthy controls) does not make any sense. So Table 5 and respective text in the results should be removed.

4.       Paragraph “blood sample processing” might be introduced in Material and methods.

English language and style need substantial improvement. Proofreading of the manuscript by native speaker is strongly recommended.

MS contains a number of typos and spelling mistakes. Using spellcheck will help you to remove most of them.

I only mention some of them

-   Please avoid using “diabetic years”, use diabetes duration

-   “glycolization” should be replaced by glycation

-   carboxyle not carboxile

-   Material not matherial

-   You use two abbeviations for fasting plasma glucose – FPG and FBG, choose one of them

-   You sometimes use taurine, sometimes taurin

-   Kruskal-Wallis not walls

-   apoptosis not apoptozis

-   vessels not wessels

and many more.

Author Response

Dear Editor,

First, we would like thank the reviewers for the helpful comments, which led us to conduct appropriate experiments.

The manuscript has subsequently been rewritten to address these concerns and comments of the reviewers.

We are grateful for your understanding and cooperation in this matter.

English

We have had our revised manuscript edited and proofread by a professional English-speaking editor (MDPI English editing) to improve

the readability and correct grammatical errors. We look forward to your reply. We believe that the language is now suitable for review.

 Response to Reviewer 2

Comments and Suggestions for Authors

In their manuscript, the authors describe results of plasma taurine measurement in a group of patients with T2DM and in control

non-diabetic subjects. They also evaluate relationship between plasma taurine and diabetic complications.

The manuscript has several flaws:

1.The description of subjects involved in the study is insufficient. Although the limitations are mentioned in the discussion, subjects’ description must be

improved. Characteristics of the subjects are shown in Table 1, however the authors pooled both groups and showed the description for all patients. So the reader

does not know which parameters differ between the groups. Furthermore, they show minimum, maximum, median, mean and standard deviation which is redundant.

I suggest using median and interquartile range as not all parameters have normal distribution. Table 1 is poorly arranged, parameters should be aligned to the left and

units should be standardized (glucose is expressed in mg/dl and taurine in mmol/l). In the abstract, the authors show median but together with SD which is incorrect.

The reference to the Table 1 belongs in “2.1 Subjects” not in the results section. Patients should be better characterised with respect to diabetic complications

(e.g. CKD stage, GFR values, DR stage etc.).

Table 1 was changed and seperated into control and diabetic groups.

Glucose levels were changed to mMol/L

We changed the Tables and numbers were corrected.

2. The authors mention lower plasma taurine in diabetic patients however this information (and respective P value) is missing in the results.

We wrote the p values and the levels of taurine.

3. Comparison of plasma taurine levels between patients with and without diabetes performed in all study subjects (including healthy controls) does not make any sense. So Table 5 and respective text in the results should be removed.

Table 5 was removed and the corrections in the text was made.

4. Paragraph “blood sample processing” might be introduced in Material and methods.

Blood sampling processing was introduced in Material and methods.

English language and style need substantial improvement. Proofreading of the manuscript by native speaker is strongly recommended.

MS contains a number of typos and spelling mistakes. Using spellcheck will help you to remove most of them.

I only mention some of them

-   Please avoid using “diabetic years”, use diabetes duration

-   “glycolization” should be replaced by glycation

-   carboxyle not carboxile

-   Material not matherial

-   You use two abbeviations for fasting plasma glucose – FPG and FBG, choose one of them

-   You sometimes use taurine, sometimes taurin

-   Kruskal-Wallis not walls

-   apoptosis not apoptozis

-   vessels not wessels

and many more.

We have had our revised manuscript edited and proofread by a professional English-speaking editor (MDPI English editing) to improve

the readability and correct grammatical errors. We look forward to your reply. We believe that the language is now suitable for review.

Round  2

Reviewer 1 Report

1. Abstract. The values in the abstract for Taurine levels are not correct. I presume the values have been interchanged. Please check.

 There is no data to support the conclusion in abstract "particularity that Taurine levels may play a role early in the disease and have no

significance in the late stage disease. I am not sure what the authors suggest by weak co-relation as the differences seam to be quite

 significant. 2. Introduction: Not sure why the authors introduce OSAS here. Numerous spell mistakes like

endotheline. 3. Results: I still wish that the author would represent the data in a visual forms preferably in box-plots. FPG or FBG. Please check.

Author Response

Dear Editor,

First, we would like thank the reviewers for the helpful comments, which led us to conduct appropriate experiments.

The manuscript has subsequently been rewritten to address these concerns and comments of the reviewers.

We are grateful for your understanding and cooperation in this matter.

English

We have had our revised manuscript edited and proofread by a professional English-speaking editor (MDPI English editing) to improve

the readability and correct grammatical errors. We look forward to your reply. We believe that the language is now suitable for review.

Comments and Suggestions for Authors

1. Abstract. The values in the abstract for Taurine levels are not correct. I presume the values have been interchanged. Please check.

There is no data to support the conclusion in abstract" particularity that Taurine levels may play a role early in the disease and have no significance in the late stage disease. I am not sure what the authors

suggest by weak correlation as the differences seam to be quite significant.

The numbers were corrected.

Table 1 showed us the significant difference of taurin levels between controls and diabetics. But, with regard to complications there was

only relation between taurin and neuropathy. So we still suggest that taurin levels may be more important for development of diabetes rather

than its complications.

2. Introduction: Not sure why the authors introduce OSAS here. Numerous spell mistakes like

endotheline. 3. Results: I still wish that the author would represent the data in visual forms preferably in box-plots. FPG or FBG. Please check.

 We removed that sentence which we did not understand how it came here. Our first manuscript did not include this sentence.

FPG was corrected as the FBG.

The box-plots were added to the result section.

Reviewer 2 Report

Dear authors,

I see that the manuscript is better now but in my opinion it should be substantially improved. I still have issues with overall presentation.  

Specific comments:

You reported that taurine levels are lower in diabetic patients but you provide median 0.8 in diabetic and 0.6 in control subjects

(in the abstract and in the results). Again (I already criticized it in my first comments) mean and SD is shown but denoted as median!

I do not see any sense in showing taurine concentration in all subjects. This sentence should be removed (in the abstract and results).

I am still missing GFR values (at least in those with diabetic nephropathy).

Two sentences that does not belong in the MS appeared in the introduction (page 2, lines 63 - 65). 

Expression "median minimum" and "median maximum" are in the first sentence of statistical analysis. You probably meant "median, minimum

and maximum".

Median and mean and SD in one sentence are shown in the results again ((page 3, line 116).

HT is in brackets although the abbreviation was introduced earlier in the text (page 3, line 117). In the same sentence and also elsewhere

in the manuscript  I would replace "comorbid disorder" with "comorbidity".

The new sentence in the result that show diabetes duration in patients with different complications is confusing as there must be patients

having two or even more complications.

Table 1 needs some improvement. You included 59 T2DM patients but you show 28 men and 12 women (and 31 men and 16 women in

the controls). If you replace min, max, mean and SD with median and quartiles you will be able to add a column with P values (you can not

express albumin as 209 +/- 444.1 - the distribution is markedly asymmetric! and not normal). I can also see FBG here but FPG in the abstract.

"our Turkey" should be replaced by "the Turkey"

Author Response

Dear Editor,

First, we would like thank the reviewers for the helpful comments, which led us to conduct appropriate experiments. The manuscript

has subsequently been rewritten to address these concerns and comments of the reviewers.

We are grateful for your understanding and cooperation in this matter.

English

We have had our revised manuscript edited and proofread by a professional English-speaking editor (MDPI English editing) to improve

the readability and correct grammatical errors. We look forward to your reply. We believe that the language is now suitable for review.

Comments and Suggestions for Authors

Dear authors,

I see that the manuscript is better now but in my opinion it should be substantially improved. I still have issues with overall presentation.  

Specific comments:

You reported that taurine levels are lower in diabetic patients but you provide median 0.8 in diabetic and 0.6 in control subjects (in the abstract

and in the results). Again (I already criticized it in my first comments) mean and SD is shown but denoted as median! I do not see any sense

in showing taurine concentration in all subjects. This sentence should be removed (in the abstract and results).

The numbers were corrected. Median was removed and replaced with mean

This sentence was removed from both abstract and result sections.

I am still missing GFR values (at least in those with diabetic nephropathy).

We measured creatinine and GFR values but as GFR values were meaningful between controls and diabetics we added it on table 1. 

Two sentences that does not belong in the MS appeared in the introduction (page 2, lines 63 - 65). 

We removed that sentence which we did not understand how it came here. Our first manuscript did not include this sentence.

Expression "median minimum" and "median maximum" are in the first sentence of statistical analysis. You probably meant "median, minimum

and maximum".

This sentence was corrected as Mean value, standard deviation, median, minimum, maximum, and frequency were used to describe the

data statistically.

Median and mean and SD in one sentence are shown in the results again ((page 3, line 116).

This sentence was corrected as mean.

HT is in brackets although the abbreviation was introduced earlier in the text (page 3, line 117). In the same sentence and also elsewhere in

the manuscript  I would replace "comorbid disorder" with "comorbidity".

Brackets was removed.

This sentence was corrected as comorbidity.

The new sentence in the result that show diabetes duration in patients with different complications is confusing as there must be patients

having two or even more complications. There were 6 patients with 2 complications, 10 patients with 10 complications, 4 patients

with 4 complications and 3 patients with 3 complications and the rest with one complication. But taurin levels were not different between them.

And we did not compare them considering duration of diabetes, so we did not mention this in the text. But duration of diabetes was